# Detection of *Leishmania* spp. in Cats: Analysis of Nasal, Oral and Conjunctival Swabs by PCR and HRM

**DOI:** 10.3390/ani13152468

**Published:** 2023-07-31

**Authors:** Maria Fernanda Alves-Martin, Thainá Valente Bertozzo, Isabella Neves Aires, Suzane Manzini, Mirian dos Santos Paixão-Marques, Lívia Maísa Guiraldi, Wesley José dos Santos, Gabriela Pacheco Sánchez, Vera Cláudia Lorenzetti Magalhães Curci, Virgínia Bodelão Richini-Pereira, Simone Baldini Lucheis

**Affiliations:** 1Department of Biology and Animal Science, School of Engineering, Sao Paulo State University (UNESP), Ilha Solteira 15385-000, SP, Brazil; 2Department of Tropical Diseases, Botucatu Medical School, Sao Paulo State University (UNESP), Botucatu 18603-560, SP, Brazil; thainabertozzo@gmail.com (T.V.B.); aires.isabella@gmail.com (I.N.A.); suzane.manzini@hotmail.com (S.M.);; 3Department of Molecular Biology and Biochemistry, School of Biological Sciences, University of California, Irvine (UCI), Irvine, CA 92697-2525, USA; pacheco.sanchez.gabriela@gmail.com; 4Biological Institute, Araçatuba Regional Laboratory, Araçatuba 16050-230, SP, Brazil; vera.magalhaes@sp.gov.br; 5Adolfo Lutz Institute, Bauru II Regional Laboratory Center, Bauru 17015-110, SP, Brazil; virichini@yahoo.com.br; 6Department of Animal Production and Preventive Veterinary Medicine, School of Veterinary Medicine and Animal Sciences, Sao Paulo State University (UNESP), Botucatu 18618-681, SP, Brazil

**Keywords:** cats, PCR, HRM, conjunctival swabs, feline leishmaniasis

## Abstract

**Simple Summary:**

Different diagnoses for Feline leishmaniasis have been studied, mainly in endemic regions of Canine Visceral Leishmaniasis (CVL), aiming to contribute to the differentiation of diseases manifested by cats with the aim of offering the best treatment for the animal. However, the study of less invasive and sensitive diagnostic techniques can be useful concomitantly with serological techniques such as the Immunofluorescent Antibody Test (IFAT) and ELISA in epidemiological surveys. In this sense, the objective of our study was to evaluate positivity for *Leishmania* from material collected in a less invasive way in serologically positive and negative cats from an endemic region for CVL; through conjunctival swabs for molecular analysis by PCR, with subsequent screening of the species by the HRM technique; and confirmation by genetic sequencing. Although few samples could be detected from the conjunctival cell samples (4/36), there was a good concordance of species confirmation by the HRM technique and genetic sequencing. Of these four animals, three were confirmed as positive by the two serological techniques, which suggests that they had contact with the parasite and produced antibodies. Therefore, HRM would be an alternative as a screening technique for different species of *Leishmania* followed by confirmation by genetic sequencing.

**Abstract:**

Background and objectives: Feline leishmaniasis (FeL) is caused by several species of parasites of the genus *Leishmania*. The disease can occur with the presence or absence of clinical signs, similar to those observed in other common infectious diseases. In endemic regions for FeL, the infection has been associated with dermatological lesions. Therefore, considering the search for less invasive and more effective diagnostic techniques, we aimed to investigate the presence of *Leishmania* spp. in domestic cats through Polymerase Chain Reaction (PCR) and high-resolution melting (HRM) analyses of conjunctival, oral, and nasal epithelial cells, and we detected the presence of anti-*Leishmania* IgG antibodies from serological techniques of the Immunofluorescent Antibody Test (IFAT) and ELISA. Methods: The PCR and HRM for detection of *Leishmania* spp. were performed on 36 samples of epithelial cells from the conjunctiva of male and female cats, collected using sterile swabs. The serological tests IFAT and ELISA were also performed. Results: The prevalence of *Leishmania donovani* infection was 11.1% (4/36) by PCR assay, and those results were confirmed for *Leishmania* species using the HRM technique. Twenty-four cats (24/36 = 66.7%) were reactive to the IFAT and twenty-two cats were reactive by the ELISA technique (22/36 = 61.1%). Interpretation and Conclusions: The use of conjunctival swabs was shown to be a non-invasive, practical, and easy-to-perform technique, and in addition to the genetic sequencing and HRM, it was able to identify the parasitic DNA of *L. donovani* in cats. This technique can be used for screening diagnosis in future epidemiological surveys of FeL and can be used as a complement to clinical and/or serological tests, as well as associating the clinical history of the animal, for the diagnostic conclusion.

## 1. Introduction

Visceral leishmaniasis (VL) is a zoonotic disease transmitted by vectors composed of more than 20 different species of the genus *Leishmania.* Currently, 92 countries are considered endemic for the disease and about one billion people live in risk areas. In addition, leishmaniasis is among the top 20 neglected tropical diseases [1].

The species responsible for the development of visceral leishmaniasis in the New World is *Leishmania chagasi* (syn. *Leishmania infantum*), while in the Old World, the species *Leishmania donovani* and *Leishmania infantum* are involved. The disease is related to a lack of infrastructure and poor hygienic conditions, being frequently reported in underdeveloped and developing countries [1].

Dogs are widely recognized as a reservoir of this disease and play an important role in the biological cycle of leishmaniasis, especially in urban and household areas; however, cases of leishmaniasis in domestic cats have also been studied in several parts of the world [1,2].

Feline leishmaniasis (FeL) in domestic cats has been reported, with the presence or absence of clinical signs, similar to those observed in other common infectious diseases in cats caused by viruses (feline leukemia virus [FeLV] and feline immunodeficiency virus [FIV]), bacteria, fungi and other protozoa. In endemic regions for FeL, infection has been associated with the presence of dermatological lesions in cats, in some cases, the only clinical sign is a lesion, with the absence of any other clinical sign [2].

Cases of feline leishmaniasis have been described in South American countries [3,4,5,6,7,8], Europe [9,10,11], Asia [12], and Africa [3].

However, although reports of leishmaniasis in cats are increasing, the role of this animal in the eco-epidemiology of the disease remains uncertain. The lack of early diagnosis of FeL in endemic areas suggests that cats continue to present potential risks of transmission of leishmaniasis to vectors since studies have shown that infection in cats can remain active for long periods; thus, the risk of infectivity to sandflies is maintained [3].

The diagnosis of feline leishmaniasis in Brazil is mostly performed through serological techniques such as the Immunofluorescent Antibody Test (IFAT) and Enzyme-Linked Immunosorbent Assay (ELISA), immunohistochemistry, and molecular tests, with emphasis on real-time PCR and conventional PCR [13,14,15]. Diagnostic techniques based on molecular assays can be applied to different types of samples, such as blood, tissue, and even epithelial cells from the conjunctiva and other mucous membranes. In the latter case, the sample is collected using sterile swabs.

The collection of material from the mucous membranes is a less invasive technique since the host’s mucosa is always in constant cell renewal and samples can be easily collected. Besides this, mucosal tissues are normally colonized by these parasites, making this type of sample favorable for parasite detection [8,16].

Therefore, considering the search for less invasive and more effective techniques, our study aimed to investigate the presence of *Leishmania* spp. in domestic cats by means of PCR on conjunctival, oral, and nasal epithelial cells that were collected with the aid of sterile swabs, and with a serological screening of cats in search of anti-IgG antibodies through the IFAT and ELISA techniques, with subsequent identification by the high-resolution melting (HRM) technique in reactive and non-reactive cats.

## 2. Material and Methods

### 2.1. Ethics Committee

This study was approved by the Ethics Committee on Animal Experimentation (CEUA) of the Botucatu Medical School (FMB-UNESP), under permit 1096/2014.

### 2.2. Collection of Samples

Epithelial cells were collected from 36 cats (both males and females) at the Zoonosis Control Center (ZCC) of Bauru, Sao Paulo State, Brazil, in the periods of October 2014 and December 2016, from all animals submitted to euthanasia during this period. These animals had no owners or were abandoned and sent to the ZCC to be euthanized.

All clinical manifestations compatible with FeL (weight loss, apathy, skin lesions, alopecia, eye infection, and lymph node enlargement) were evaluated and recorded in individual files for each animal. Different sterile swabs were used to collect the epithelial cell material, to remove eye discharge from the conjunctiva, and to collect oral and nasal mucosae. The samples were stored in sterile microtubes, which were kept under refrigeration at 4 °C until the time of analysis. 

A collection of 5 mL of blood from each animal was also carried out to perform the serological screening using the IFAT and ELISA techniques. The blood samples were placed in a Styrofoam box under refrigeration until arrival at the laboratory and then kept at −20 °C until the moment of performing the tests.

### 2.3. Immunofluorescent Antibody Test—IFAT

The IFAT technique was performed according to CAMARGO [17]. On the slides fixed with the *L. infantum* antigen, 10 µL of diluted serum (1:40) was distributed, incubating at 37 °C for 30 min in a humid chamber. They were then washed twice with 0.01 M PBS (Phosphate buffered saline) pH 7.2 for 10 min, then dried in an oven. The species-specific conjugate (Anticat IgG FITC, Sigma^®^, St. Louis, MO, USA) was diluted according to a pre-established titer in a 20% Evans Blue solution, in which it was previously diluted in 0.01 M PBS pH 7.2 in the proportion from 1:5. Then, 10 µL of the conjugate was distributed for each dilution, incubating again at 37 °C for 30 min in a humid chamber. The washings were carried out again twice with PBS 0.01 M pH 7.2 for 10 min and then dried in an oven. The slides were mounted with buffered glycerin pH 8.5, covered with 24 × 60 mm coverslips, examined under an Immunofluorescence microscope with a 400× objective. After reading the controls, read the serum samples from the animals, serum samples with a title equal to or greater than 40 being considered positive. Positive sera in the 1:40 dilution were evaluated in higher dilutions (1:40; 1:80, 1:160, 1:320, and 1:640) to determine the final titer of anti-*Leishmania* IgG antibodies in the cats.

### 2.4. Indirect ELISA Technique

The indirect ELISA test was performed according to the technique described by Costa et al. [18] for *Leishmania* spp. For the ELISA test, in each well of the polystyrene plate (Nunc^®^ Maxisorp Thermo Scientific, Waltham, MA, USA), 100 µL of soluble *Leishmania infantum* antigen at a concentration of 10 µg/mL was diluted in 0.05 M sodium carbonate-bicarbonate, pH 9.6. After incubation of the plate for 18 h in a humid chamber at 4 °C, the excess antigen was removed by four consecutive washes, with PBS 0.01M, pH 7.4, containing 0.05% Tween 20 (PBS-T). The plates were blocked with PBS pH 7.4 plus 1% skimmed milk powder, in a humid chamber, at 37 °C, for two hours. After four washes with PBS-T, to remove the blocking solution, 100 µL was added per well, in duplicate, from serum samples—from the positive control animal, negative control, and from the cats—to be tested at a 1:400 dilution in PBS-T plus 10% fetal bovine serum. Plates were incubated for one hour at room temperature and then washed four times with PBS-T. Then, 100 µL of anti-total cat IgG conjugate linked to peroxidase (A20-120P; Bethyl, Montgomery, TX, USA) at a 1:40,000 dilution in PBS-T was added to each well of the plate, followed by a new incubation in a humid chamber at 37 °C for 45 min. After washing, 100 µL of tetramethylbenzidine dihydrochlorinated (TMB) solution (BD Biosciences Pharmingen, San Diego, CA, USA) was added to each well of the plate, with subsequent incubation of the plate for 30 min protected from light at room temperature. The reaction was interrupted by blocking each well with 50 µL of 0.5 N sulfuric acid and the optical density (O.D.) was determined in an ELISA reader (Universal Microplate Reader—EL 800—BIO-TEK Instruments, Inc., Winooski, VT, USA), using 450 nm filter.

### 2.5. DNA Extraction 

DNA extraction from the swabs was performed in accordance with Lahiri and Nurnberger [19], with modifications as described by Oliveira et al. [8].

### 2.6. Polymerase Chain Reaction and Sequencing

Two different PCR protocols were performed using two different pairs of primers, both referring to trypanosomatid DNA detection.

Firstly, the primers LITSR and L5-8S targeting the ITS1 gene as described by El Tai et al. [20] were used. The resulting amplicons varied from 300 to 350 base pairs, depending on isolated *Leishmania* species. Subsequently, for amplification of the gene encoding of the heat shock protein (HSP70), the primers HSP70F and HSP70R were used, as described by Hernández et al. [21]. Strains of *L. infantum* (MHOM/BR/IOC/2906) were used as positive controls for each PCR reaction and ultrapure water was used as negative control.

The PCR results were confirmed using the DNA sequencing method. The PCR products were purified using Agencourt AMPure XP (Beckman Coulter, Inc., Brea, CA, USA) and were sequenced using the BigDye^®^ Terminator v.3.1 cycle sequencing kit (Applied Biosystems, Foster City, CA, USA) and automatic sequencing in an ABI3500 Genetic Analyzer (Applied Biosystems). Then, the sequences were analyzed using the data from GenBank and also the basic local alignment search tool (BLAST) software (http://blast.ncbi.nlm.nih.gov). 

### 2.7. High-Resolution Melting (HRM) 

For identification of parasitic species detected in cats, the HRM technique was performed using the Master Mix MeltDoctor HRM (Applied Biosystems^®^, Inc., CA, USA) according to Hernández et al. [21].

## 3. Results

Of the 36 sera from animals submitted to screening, 24 were reactive to the IFAT (24/36—66.7%) and 22 sera from animals were reactive to the ELISA technique (22/36—61.1%). The results of the IFAT and ELISA serological tests showed good agreement and accuracy with each other (K = 0.64). There was discordance in only four samples that were reactive only by the IFAT and two samples that were reactive only by ELISA.

Through molecular detection of epithelial cells from the conjunctiva utilizing primers ITS1 and HSP70 collected using sterile swabs, positive DNA samples were obtained from four animals (4/36; 11.1%). Three out of four animals had clinical manifestations and positive serologies (Table 1). There was no positivity from the swabs collected from the oral and nasal mucosae. *Leishmania* isolates from two cats in our study were shown to have 98% similarity with *L. donovani* strains in the GenBank (Number JQ 780821.1). By the analysis of alignment temperatures from melting curves of the swab samples, using the HRM technique, the parasite species in the two samples was also identified as *L. donovani* (Figure 1). 

One of the other two samples tested showed 90% similarity to *Trypanosoma theileri* (*T. theileri*) strains in GenBank (Number AB56920.1).

## 4. Discussion

The main question regarding infections in cats is based on the fact that they could be less susceptible to a disease or could they be more resistant to infection. Studies show that susceptibility increases across the animal’s lifetime, assuming that the exposure time is related to the chances of infection [22] (Pennisi et al., 2012). In addition, the natural resistance of felines may explain the different infection prevalence rates in this species, which requires further study to better assess the immune response of cats infected with *Leishmania* [5,23] (Solano-Gallego et al., 2007; Silveira-Neto et al., 2015).

In our study, PCRs were performed on samples of swabs from the conjunctiva and the oral and nasal mucosae. Only swabs from conjunctiva were positive for parasites of four cats. Nasal and oral swab samples were believed to have a high potential for qualitative molecular diagnosing of canine visceral leishmaniasis since the results were equivalent to those observed from samples collected invasively [15].

Analyzing the four positive animals for PCR of conjunctival swabs, one of them was not reactive to any serological technique, which suggests that this animal came into contact with the parasite recently and still did not produce antibodies to fight the infection, or that the non-production of antibodies occurred due to their weakened immune status. As for the other three animals that were serologically positive, it is suggested that they came into contact with the parasite, produced antibodies, and manifested the infection or not. The detection of the parasite by the molecular technique may be due to an exacerbation caused by possible immunosuppression, the occurrence of another infection, or even due to reinfection. Thus, we must pay attention to the presence of clinical signs in cats, always associating the diagnosis with epidemiology, especially when the animal comes from an endemic area for visceral leishmaniasis.

The use of less invasive techniques to collect biological material from cats, such as swabs from the conjunctiva and mucosa, may be a good screening alternative in epidemiological studies. Some researchers have used this technique for diagnosing FeL and reported that swab collection was more practical, easier, and less invasive, which confirms the potential usefulness of this type of sample collection in studies of cats [8].

The results of this present study were similar to those reported in Brazil [8], which detected cats infected with *Leishmania* spp. from swab samples taken from the conjunctiva of 52 cats, with positivity rates of 28.6% (2/7) and 11.1% (5/45) among cats in Pirassununga, Sao Paulo, and Ilha Solteira, Sao Paulo, respectively.

Another study carried out in Greece with PCR evaluations of swab samples, found lower diagnostic values and lower sensitivity in comparison with skin biopsy and bone marrow samples. Nonetheless, the importance of using different techniques for diagnosing FeL was emphasized [24]. 

In a study developed by Morelli et al. [25], a statistical association was verified between anti-*L. infantum* antibodies and cohabitation with dogs, indicating that feline populations living in the examined Italian and Greek touristic areas were exposed to *L. infantum* and that they may contribute to the circulation of this parasite, enhancing the risk of infection for dogs and humans.

The diagnosis of *Leishmania* spp. was verified at high rates in dogs and cats from central Israel in a study of *L. tropica*. The prevalence of infection by kDNA PCR for *Leishmania* spp. was 22.8% for dogs and 28.9% for cats, and the results showed that *Leishmania* infection is widespread in both the feline and canine populations in a focus of cutaneous human leishmaniasis in the central area of Israel [26].

Although no studies of cats have yet reported the degree of conjunctival and mucosal parasitism, or whether positive findings from swab samples via PCR are related to the animal’s immune status, we observed that three of the four positive animals presented clinical manifestations. This shows that the animal’s immune status is not always related to the presence of the parasite in mucosal cells and consequent molecular detection. Detection of *Leishmania* spp. infection in cats, in combination with the absence of clinical signs, might indicate that these animals are asymptomatic carriers of these parasites [8]. However, clinical manifestations are not sufficient to diagnose *Leishmania* infection nor other clinical pathologies that can have similar manifestations or symptoms since in our study one of the animals reactive to PCR and serology did not show clinical signs compatible with FeL, while another serologically negative animal showed compatible clinical signs. Thus, it is suggested that only clinical and/or serological tests are not enough to conclude the diagnosis; the epidemiological context of the animal must be considered for the conclusion of the case.

Identification by genetic sequencing of *L.donovani* in two samples of swabs was confirmed by the HRM technique. The other sequenced sample identified the DNA of *T. theileri* in cats, an uncommon parasite in this species since it is a trypanosomatid commonly detected in cattle and transmitted by hematophagous flies; it is believed that the contact of these cats with these vectors, either by bite or even by ingestion of the insect, has contributed to the infection of these felines.

In our study, we used the HRM technique for the analysis of thermokinetic differences between amplicons that occur due to differences in the composition of the nucleotides between them. Therefore, HRM is able to differentiate the existing polymorphisms between the nucleotides [27]. The use of HRM has been showing good performance and reliability; moreover, it is a fast, simple, and low-cost technique, with great applicability in endemic regions [21].

From the HRM, we identified the DNA of *L. donovani* in two samples of conjunctival swabs; therefore, it was possible to verify the effectiveness of this technique as an alternative for screening positive samples from trypanosomatid primers with subsequent genetic sequencing for the determination of the parasitic species. This efficacy was confirmed in studies that allowed discriminating Brazilian, European, and African species of *Leishmania*, with high sensitivity and precision, showing the ability to detect less than one parasite per reaction, even in the presence of host DNA [28,29].

Thus, with such results, we can suggest the use of HRM to confirm the parasitic species in FeL for its effectiveness, practicality, and speed of obtaining results. The use of this technique, together with a molecular diagnosis, can confirm the presence of the parasite in different biological samples and collaborate in epidemiological surveys.

## 5. Conclusions

The use of swab samples from the conjunctiva was shown to be a non-invasive, practical, and easy-to-perform technique in cats. Together with genetic sequencing and HRM, this was able to identify parasite DNA from *L. donovani* in the species. This technique can be used for diagnostic screening in future epidemiological surveys, associated with serological techniques. In addition, it is necessary to consider the epidemiological context and the clinical evaluation of the animals, for better diagnostic coverage.

## Figures and Tables

**Figure 1 animals-13-02468-f001:**
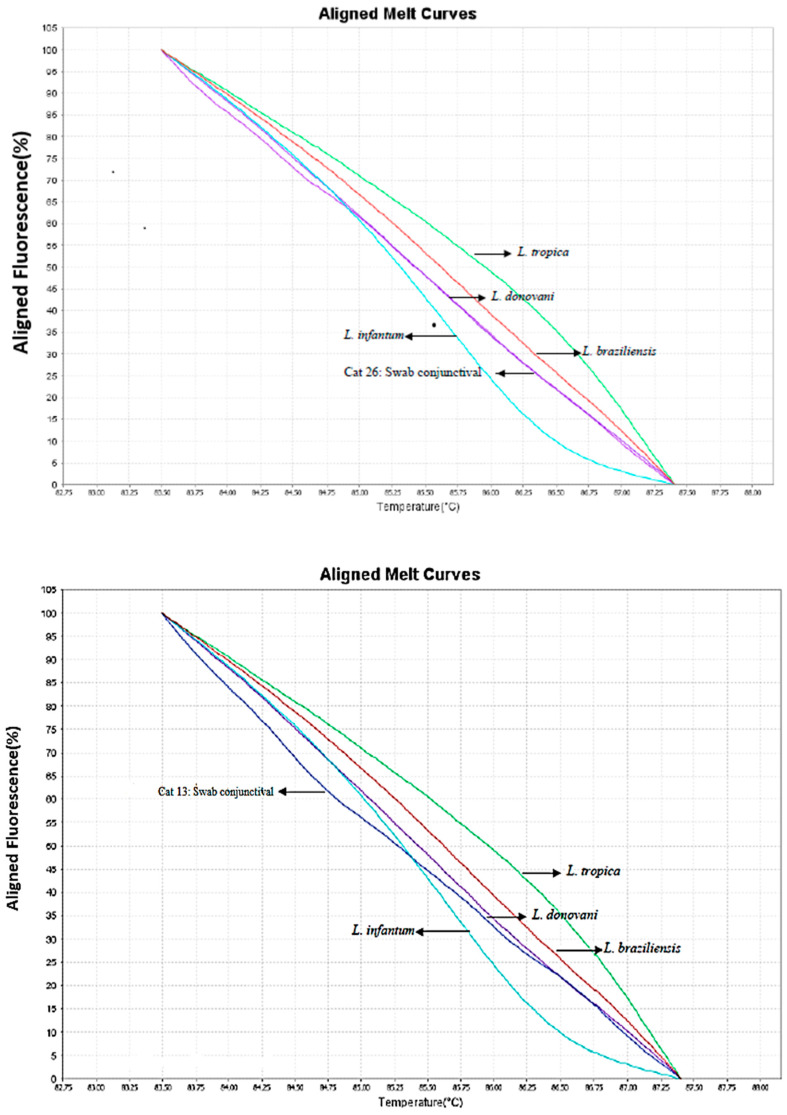
Representation of curve profiles with standard strains of *Leishmania* and positive samples of conjunctival swabs, using primer HSP70.

**Table 1 animals-13-02468-t001:** Serological results and molecular detection of *Leishmania* spp. from the conjunctival, oral, and nasal swabs in domestic cats.

				Polymerase Chain Reaction—PCR ITS1 and HSP70 Genes				
Identification	Clinical Signs	IFAT	ELISA	Conjunctival Swab	Oral Swab	Nasal Swab	Sequencing/ Accession Number	High-Resolution Melting—HRM
Cat 10	Skin lesions	Negative	Negative	Positive	Negative	Negative	-	-
Cat 13	Weight loss, apathy, skin lesions, alopecia, eye infection, and lymph node enlargement	Positive	Positive	Positive	Negative	Negative	*L. donovani* (98%)/JQ780821.1	*L. donovani*
Cat 15	No signs	Positive	Positive	Positive	Negative	Negative	*T. theileri* (90%)/AB56920.1	-
Cat 26	Weight loss, skin lesions, and alopecia	Positive	Positive	Positive	Negative	Negative	*L. donovani* (98%)/JQ780821.1	*L. donovani*

## Data Availability

The datasets used and/or analyzed during the current study are available from the corresponding author upon reasonable request.

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
