# Peer review of "Detection of Leishmania spp. in Cats: Analysis of Nasal, Oral and Conjunctival Swabs by PCR and HRM"

_animals, 2023, doi:10.3390/ani13152468_

Round 1

Reviewer 1 Report

I was invited to review your manuscript entitled “Detection of Leishmania spp. in cats: analysis of nasal, oral and conjunctival swabs by PCR and HRM”, which describes a study on Leishmania detection in domestic cats through different molecular procedures starting from different swabs (i.e., oral, nasal and conjunctival swabs). Different molecular approaches were tested targeting ITS1 and HSP70 genes and real-time HRM PCR. The Manuscript is generally linear but it needs to be revised in some parts. My suggestions are reported in the attached doc file

Reviewer 2 Report

Major comments

The manuscript is interesting, but it is disappointing that it contains very few positive samples to confirm the reliability of the HRM technique from swabs. Could other samples have been taken in other facilities or from live cats? However, it’s a fact that the low rate of positive samples for feline leishmaniasis seems common in the literature.

In the results, line 121, the authors indicate that they have sequences for 2 cats out of 4. It would be interesting to know why sequences for the other cats are missing.

Figure 1: the title and eventually a legend are missing. The abscissa and ordinate data are not very readable.
It can be seen that other Leishmania species were used in the study (HRM part). These samples are not mentioned in the materials and methods section, nor is it indicated how they were processed. It would be useful to add this information to the materials and methods.
Would it be easier to discriminate (visually) between species if the curves of the HRM results were presented in derivative form?

Minor comments

Throughout the manuscript, the authors refer to PCR; it would be appreciated to indicate that this refers (also) to sequencing.

 In the sentence lines 49-50 "infection has been associated with the presence of dermatological lesions in cats" seems to say that all infections are characterized by lesions. Wouldn't it be rather that : in some cases, the only clinical sign is a lesion, in the absence of any other clinical sign? Knowing that asymptomatic cats can harbor the parasite.

Perhaps describe the material and method a little more, depending on the number of words allowed:

As the extraction protocol is indeed long, perhaps just indicate the modification made in this protocol (even if the bibliographic reference is given for this modification).

For PCR, it would be appreciated to indicate :
- the final volume of the mix PCR.
- sequences of primers used (even if the bibliographic reference is given)
- PCR cycles used
- whether the amplification products were visualized on gel or otherwise before sequencing.

Line 98, the names of the primers used are indicated, perhaps also indicating that this is the ITS1 region.

For HSP70 primers, the size of the amplicons should be indicated.

Line 102 states that primers are used as described in reference no. 19. Does this refer to the PCR mix, the PCR cycle or just the primer sequences?

For HRM, the primers used should be indicated and whether the PCR mix and PCR program are the same as those used for PCR above (or as in reference 19).

Was an internal control used to check that negative samples were not the result of the presence of inhibitors or of excessively degraded DNA?

There are a few typos that need to be corrected (line 20, 60, 66, 121), the table title should be above the table, the internet address of reference 1 is invalid.

The meaning of certain acronyms is missing: line 59 (IFAT and ELISA), line 78 (SP), eventually HRM line 71 even if already indicated in the abstract.

Round 2

Reviewer 1 Report

Dear Authors, 

I thank you for your detailed reply. I think that your study needs to be better highlighted. You had collected much more data than you described in the Manuscript. I believe that the readers need to know all the steps of your study including 1) serological tests, 2) their results, 3) serological/PCR comparison, 4) serological results in Table 1, 5) sequencing of Trypanosoma theileri in Table 1,  and 6) the effectiveness of your PCR protocols in identifying Leishmania genus and other trypanosonomatid genus. Thus, I suggested you add the content of your replies (line 77, line 83, line 113, line 114, line 123, lines 124-126, line 169-171) in the appropriate paragraphs along the Manuscript (don't worry about the characters' numbers). Moreover, consider reviewing accordingly the discussion and the conclusions adding considerations on serological results and PCR comparison.

Some minor revisions:

- lines 98, 105, 151: delete the year of publications, not necessary in this type of references list

- line 167: Leishmania in italics
